# STOPS approach to individualised physiotherapy versus usual physiotherapy care for chronic low back pain in India: A randomised controlled trial protocol

Andrew J. Hahne[1]*, Mohini Shah[1,2], Muhammed Rashid[1,2], Kavitha Raja[2], Musa Sani Danazumi[1], Jon Ford[1,3]

1 Discipline of Physiotherapy, School of Allied Health, Human Services & Sport, La Trobe University, Melbourne, Victoria, Australia, 2 JSS College of Physiotherapy, Mysuru, Karnataka, India, 3 Advance Healthcare, Boronia, Australia

* a.hahne@latrobe.edu.au

## Abstract

### Background

Low back pain (LBP) is the leading cause of disability worldwide, particularly in low- and middle-income countries such as India. Many treatment approaches fail to address the multidimensional nature of LBP, leading to suboptimal outcomes. The Specific Treatment of Problems of the Spine (STOPS) approach addresses biological, neurophysiological, and psychosocial factors to deliver individualised physiotherapy for LBP, yet its effectiveness in India has not been explored.

### Objective

This study aims to evaluate the effectiveness of individualised physiotherapy using the STOPS approach compared to usual physiotherapy care in individuals with chronic low back pain (CLBP) in India.

### Methods

This is a parallel-group superiority randomised controlled trial with blinding of participants, outcome assessors, and the data analyst. A total of 154 participants in India with CLBP will be recruited and randomised to receive 11 sessions of either individualised physiotherapy via the STOPS approach or usual physiotherapy care. The primary outcome is activity limitation measured using the Oswestry Disability Index at 26 weeks. Functional MRI and qualitative interviews will assess brain functional changes and participant experiences, respectively. Data will be analysed using intention-to-treat principles.

**Data availability statement:** No datasets were generated or analysed during the current study. All relevant data from this study will be made available upon study completion.

**Funding:** JSS College of Physiotherapy funded the fMRI scans and physiotherapy treatment.

**Competing interests:** The authors have declared that no competing interests exist.

## Conclusion

This study will provide insights into the effectiveness of the STOPS approach to delivering individualised physiotherapy for CLBP in India, in comparison to usual physiotherapy care.

## Trial registration

This trial is prospectively registered with the Clinical Trials Registry of India: CTRI/2024/08/072259, https://ctri.nic.in/Clinicaltrials/pmaindet2.php?EncHid=MTAyNjY2&Enc=&userName=2024/08/072259

## Introduction

Low back pain (LBP) has a considerable impact on people's lives and the global economy [1]. It is estimated that 619 million people experienced LBP globally in 2020, making this condition the leading cause of disability worldwide [1]. South Asia is home to one of the highest number of people with LBP (117 million) among all regions of the world, accounting for approximately 19% of the years lived with disability due to LBP globally [1]. It is estimated that the number of people with LBP will increase by 36.4% globally by 2050 [1], especially in low-income and middle-income countries [1,2].

Although many episodes of acute LBP (< 6 week duration) resolve quickly [3], systematic reviews have shown high rates of persistent or recurrent pain after 12 months [3,4]. Chronic low back pain (CLBP), defined as pain persisting for > 3 months, accounts for most of the healthcare costs and economic burden, making this condition a key research priority [5].

The underlying mechanisms driving CLBP are complicated, involving multiple potential biological, neurophysiological, and psychosocial factors [6–9]. In this context, treatment approaches that target a single mechanism are unlikely to be effective for a high proportion of people with CLBP [7,10]. There is growing evidence to support individualised approaches that tailor treatment for each person with CLBP based on biopsychosocial mechanisms [7,11].

The Specific Treatment of Problems of the Spine (STOPS) is an individualised physiotherapy approach for LBP where a treatment program is tailored for each patient based on known and hypothesised biological, neurophysiological and psychosocial mechanisms [12]. The STOPS approach was developed and tested in Australia, where it has demonstrated effectiveness and cost-effectiveness in a randomised controlled trial (RCT) [7,13,14]. STOPS has not yet been implemented or evaluated in other countries.

An opportunity exists to evaluate the effectiveness of the STOPS approach to individualised physiotherapy versus usual care in a low- and middle-income country (LMIC) such as India. Surveys in Nigeria [15] and India (article in preparation) demonstrate very low recognition of the STOPS approach among physiotherapists in such countries, providing fertile ground for an implementation trial. In addition,

physiotherapy is a common treatment option for CLBP in LMICs, but typically involves approaches with low evidence of effectiveness such as electrotherapy modalities [15,16]. Combined with high CLBP prevalence rates in LMICs, implementing and establishing the effectiveness of STOPS in these countries has potential to significantly reduce the burden of CLBP.

This paper reports a protocol for a RCT comparing individualised physiotherapy based on the STOPS approach versus usual physiotherapy care for people with CLBP in India. It is hypothesised that individualised physiotherapy (STOPS) will produce superior outcomes to usual physiotherapy care.

## Methods and materials

### Study design

This participant, outcome assessor, and data analyst-blinded, parallel-group superiority RCT will examine the effectiveness of individualised physiotherapy based on the STOPS approach versus standard physiotherapy (usual care) for the management of CLBP in India.

### Registration and compliance

This study was prospectively registered with the Clinical Trials Registry of India (CTRI/2024/08/072259, https://ctri.nic.in/Clinicaltrials/pmaindet2.php?EncHid=MTAyNjY2&Enc=&userName=2024/08/072259). The study protocol (S1 File) was reviewed and approved by the institutional ethical committee of JSS Academy of Higher Education and Research, Mysuru, India (JSSMC/IEC/27102023/25 NCT/2023–24). Protocol changes will be communicated to the ethics committee after review by all members of the research team. The SPIRIT checklist (S2 File), and SPIRIT schedule of enrolment, interventions and assessments (Fig 1) are provided.

### Setting

The study will be conducted at the Physiotherapy Outpatient Department of JSS Hospital, Mysuru, India, and its associated community centres. The hospital has a capacity of 2000 beds and a community outreach program that covers 5000 patients per year.

### Public involvement

A qualitative study exploring perceptions of LBP and physiotherapy treatment was conducted in the local area where the trial will run (manuscript in preparation). This helped to inform trial design and procedures for the RCT.

### Recruitment

Patients with CLBP seeking physiotherapy will be invited to participate in this study. The recruitment will be supplemented by targeted social media advertisements and public announcements. Individuals who express interest in the study will be given detailed study information by a trial physiotherapist, with written informed consent obtained.

### Participants

Table 1 outlines the participant selection criteria.

### Treating physiotherapists

Qualified physiotherapists practicing in India at JSS Hospital will treat participants in the usual care group. Two registered physiotherapists in India, who have completed comprehensive training and mentoring in the STOPS approach led by the original Australian developers (JF, AJH), will treat the patients in the individualised physiotherapy group.

| Study period | | | | | | | |
|---|---|---|---|---|---|---|---|
| | Enrolment | Allocation | Post-allocation | | | | Close-out |
| Timepoint | -t1 | 0 | t1 (baseline) | t2 (5 weeks) | t3 (10 weeks) | t4 (26 weeks) | t5 (52 weeks) |
| **ENROLMENT:** | | | | | | | |
| Eligibility screening | X | | | | | | |
| Informed consent | X | | | | | | |
| Allocation | | X | | | | | |
| **INTERVENTIONS:** | | | | | | | |
| Individualised physiotherapy (STOPS) | | | ◄─────────────────── ▶ | | | | |
| Usual physiotherapy care | | | ◄─────────────────── ▶ | | | | |
| **ASSESSMENTS:** | | | | | | | |
| Primary outcome (Oswestry) | | | X | X | X | X | X |
| Secondary outcomes | | | X | X | X | X | |
| fMRI (n=30) | | | X | | X | | |
| Patient interviews (n=40) | | | | | X | | |

**Fig 1. SPIRIT schedule of enrolment, interventions and assessments.**

Physiotherapists assigned to treat the individualised physiotherapy group prepared for training firstly by reading research studies on the STOPS approach [13,21–24]. Additionally, they completed a comprehensive online STOPS training course comprising four modules, and they watched video tutorials demonstrating LBP assessment and treatment techniques. They also practised patient assessments using a standardised template.

Following the completion of pre-learning, the physiotherapists assigned to treat the individualised physiotherapy group completed a nine-week internship in Australia, where they took part in training. They interacted with other PhD students working on STOPS-related projects through peer learning, and were mentored and trained by the original developers of the STOPS approach (JF, AJH). During this period, they also reviewed real patient assessment examples and took part in role-play exercises, where they practised assessing each other by simulating real patient scenarios. Additionally, they recorded video demonstrations of important assessment and treatment techniques. These recordings were reviewed by peers and STOPS mentors, with constructive feedback provided to help improve their clinical skills. These recordings were practised and refined until the expected level of competency was reached.

The training was delivered through a mix of online sessions and in-person tutorials, with the former focusing on clinical reasoning and the latter on practical skill development. Moreover, the STOPS physiotherapists viewed video examples in which one of the original developers (JF) explained the main treatment principles to patients. After this, they recorded their own explanations of the same content, which were reviewed by their peers and mentors (JF and AJH) to check for clarity and effectiveness. The training concluded with a practical examination designed to assess proficiency in applying the STOPS protocols. Successful completion of this assessment was required prior to commencing patient treatment in the trial.

**Table 1. Eligibility criteria common to all subgroups.**

| Inclusion criteria |
| --- |

1.  A primary complaint of either:
a. low back pain, defined as pain between the inferior costal margin and the inferior gluteal fold with or without referral into the leg(s) [17,18],
or
b. referred leg pain without back pain, defined as predominately unilateral posterior leg pain extending below the knee, or anterior thigh pain, presumed to be of lumbar spine origin.
2. Duration of the current episode of primary complaint lasting for greater than 3 months (chronic stage of the injury) [19]. A new episode is distinguished from a previous episode by at least a 4-week pain-free period [20].
3. Aged between 18 and 65 (inclusive)
4. Fluency in English or Kannada language, sufficient to complete questionnaires and to enable understanding of the intervention which can be administered in either of these languages
5. Agreeing to refrain from other interventions wherever possible for the 10-week main treatment period of the trial, aside from consultations with medical practitioners, medication, and any exercises already being undertaken.

| Exclusion criteria |
| --- |

1. Active cancer under current treatment, as the treatment of cancer may interfere with their ability to participate in the trial
2. Signs of cauda equina syndrome based on bladder or bowel disturbance and/or imaging
3. Current pregnancy, or childbirth within the last 6 months
4. Spinal injections within the last 6 weeks, as we wish to study treatment effects independent of the effects of injections
5. Any history of lumbar spine surgery
6. A pain intensity score of less than 2/10 on a 0–10 numerical rating scale due to low severity
7. Ability to walk, sit, and stand for one hour or more and no sleep disturbance at night, as we wish to exclude people with low severity
8. Already received more than 5 sessions of physiotherapy with any of the treating physiotherapists before enrolment
9. Inability to walk safely, such as severe foot drop causing regular tripping
10. Planned absence of more than one week during the treatment period (such as overseas holidays)

## Randomisation and allocation concealment

Eligible participants will be randomly assigned to receive either individualised physiotherapy or usual physiotherapy care. A researcher from Australia not involved in screening, enrolling, or treating participants will use web-based randomisation software to generate a randomisation sequence. Block randomisation with random block lengths will be used to ensure relatively consistent distribution of patients between the two groups throughout the trial.

The randomisation sequence will be sent to a research clerk in India who will have no involvement in screening, enrolling, or treating participants. The clerk will prepare 154 envelopes. A card labelled either "STOPS" or "UC" will be sealed within opaque envelopes, numbered consecutively, in accordance with the randomisation sequence. The treating physiotherapists will ensure that the envelopes are opened sequentially in numerical order when each participant is entered into the trial.

## Individualised baseline assessment

Information from the initial eligibility screening and physical examination will be used to confirm eligibility and provide descriptive information on the baseline characteristics of the patients. An individualised assessment of each patient at baseline will be completed by the assessing physiotherapist to establish the following elements critical for directing individualised care (Fig 2):

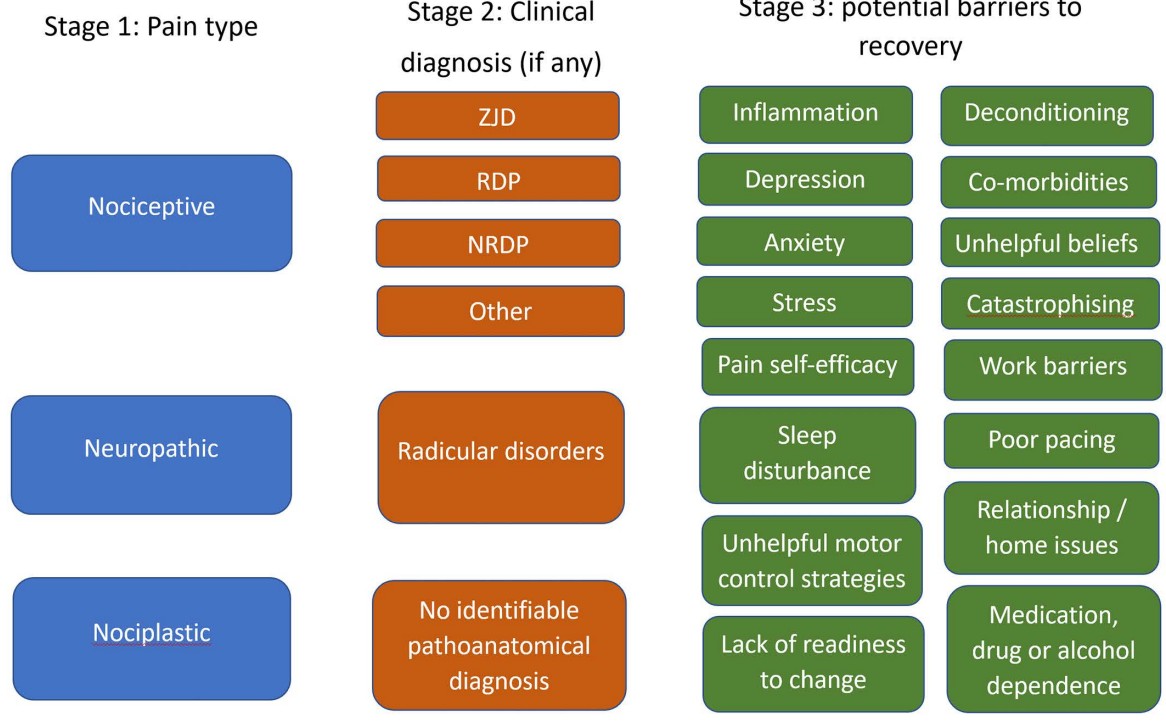

ZJD = zygapophyseal joint dysfunction, RDP = reducible discogenic pain, NRDP = non-reducible discogenic pain

**Fig 2. Pain types, subgroups, and biopsychosocial barriers to recovery evaluated to guide individualised physiotherapy.**

i. Predominant pain mechanism (nociplastic, neuropathic, or nociceptive) based on established and validated assessment criteria [25–29].

ii. Classification into a LBP diagnostic subgroup, adapted from the original STOPS protocols [21–24,12] as detailed in S3 File. The subgroups will be: i) zygapophyseal joint dysfunction, ii) reducible discogenic pain, iii) non-reducible discogenic pain, iv) radicular disorder, v) "other" (e.g., sacroiliac joint dysfunction), and, vi) "no identifiable pathoanatomical diagnosis".

iii. Evaluation of additional biopsychosocial barriers to recovery, based on clinical assessment and standardised screening tools. These include tools to detect inflammatory processes [30], poor prognosis [31], symptoms of depression, anxiety, and stress [32], pain self-efficacy [33], pain catastrophising [34], insomnia [35], and high levels of disability [36].

### Interventions

All participants will receive 10 x 45-minute physiotherapy sessions throughout a 10-week period, matching the period of intervention in the successful STOPS trial in Australia [13]. An 11th session will occur 6-months post randomisation to review, reinforce and progress the treatment strategies. Any ongoing care after the 6-month review will be organised by the treating physiotherapist. A Template for Intervention Description and Replication (TIDieR) table [37] is presented in Table 2 to summarise the key elements of the treatment protocol for each group.

**Table 2. Description of experimental and comparison group treatments according to the template for intervention description and replication (TIDieR).**

| | Individualised physiotherapy | Usual physiotherapy care |
|---|---|---|
| Brief name | STOPS | Usual care |
| Why | Improve pain and activity limitation by addressing each participant's individual barriers to recovery, with better maintenance of treatment effects in the long-term | Short-term relief of pain and function through traditional treatment methods that meet the expectations of participants |
| What materials | STOPS treatment protocols. Treatment based on thorough assessment and questionnaire battery. | Usual care treatment as per protocols followed in Physiotherapy Outpatient Department (OPD) of JSS Hospital, Mysuru, India. |
| What procedures | In accordance with published STOPS protocols, modified for persistent pain. | Usual treatment procedures at discretion of physiotherapist |
| Who provided | Physiotherapists who have received training in the STOPS approach for managing persistent low back pain | Physiotherapists unfamiliar with the STOPS approach |
| How provided | Individual face-to-face sessions | Individual face-to-face sessions |
| Where (setting) | Participant with chronic LBP living independently in the community. Treatment provided at Physiotherapy Outpatient Department (OPD) of JSS Hospital, Mysuru, India and the associated community centres affiliated to JSS Hospital. | Participant with chronic LBP living independently in the community. Physiotherapy Outpatient Department (OPD) of JSS Hospital, Mysuru, India and the associated community centers affiliated to JSS Hospital. |
| When/how much (dose) | 11 x 45-minute sessions (10 sessions in 10 weeks, and the 11th at 6-months post randomisation | 11 x 45-minute sessions (10 sessions in 10 weeks, and the 11th at 6-months post randomisation |
| Tailoring | Sessions tailored to the needs and progress of the individual in accordance with the STOPS protocols and detailed assessment profile of the participant | Sessions tailored to the needs and progress of the individual based on standard procedures used by the physiotherapist |
| Fidelity checking measures | Physiotherapists must pass a written and practical examination prior to commencing treatment in the trial. A minimum of one treatment session per month will be video recorded and assessed by the researchers for compliance with treatment protocols. Individual patient notes will be reviewed twice throughout the treatment program for each participant by the researchers, to evaluate fidelity and to provide timely feedback | Treatment will be recorded to determine its contents. |

**Individualised physiotherapy group.** Treatment will follow the core principles established in the treatment protocols for the original STOPS trial [13,21–24,12]. Physiotherapists will have several treatment options to choose from, depending on the participant's initial assessment findings and their response to treatment evaluated via regular reassessments. STOPS treatment approaches are listed in Table 3 and explained in detail in S4 File.

**Usual care group.** Treatment provided to participants in this group will be at the discretion of their treating physiotherapist in accordance with their standard practice. These physiotherapists will not receive training in the STOPS approach. Specific treatments administered will be documented for each participant and reported with trial results.

## Blinding

Participants will be blinded to the study hypothesis, and they will not be told what treatments are being compared in the trial. Participants will not be informed that there are two different treatment groups in the trial. They will only be

**Table 3. Available treatment strategies for the individualised physiotherapy group.**

1. Advice and education
2. Goal setting
3. Pacing
4. Management of inflammation
5. Sleep management
6. Motor control training
7. Management of psychosocial barriers to recovery
8. Targeted manual therapy techniques
9. Directional preference management
10. Graded exercise
11. Graded activity
12. Pain management strategies
13. Mindfulness strategies
14. Management of increases in pain
15. Referral to other healthcare providers

told that they will receive treatment based on the clinical judgement of their treating physiotherapist. Unblinding will be available for significant adverse events in consultation with the ethics committee. Outcome assessment will be via patient questionnaires, and will not involve treating physiotherapists. The data analyst will be provided with grouped data without identifying treatment group labels. It will not be possible for treating physiotherapists to be blinded to treatment allocation.

## Treatment fidelity

Physiotherapists delivering individualised physiotherapy will be closely monitored using the following methods to evaluate and promote treatment fidelity:

- Completed baseline assessment forms, including subgroup classification, identified pain type, and prioritised barriers to recovery, will be reviewed by STOPS mentors (JF and AJH) for confirmation and constructive feedback.

- Weekly online tutorials will be held during the initial 3 months of the trial. These sessions will involve discussions between the STOPS physiotherapists and mentors (JF and AJH) to review baseline data, clinical reasoning forms, and treatment records. The frequency of these tutorials will reduce in the later part of the trial as treating physiotherapists become more independent.

- Treating physiotherapists' clinical notes from a random selection of participants will be evaluated during the treatment period, with group and individual feedback provided by STOPS mentors (JF and AJH).

## Outcome measures

Outcomes will be assessed using self-administered patient questionnaires that will be filled in physically at the clinic at baseline, and at the 5-week, 10-week and 26-week treatment sessions. The primary outcome will also be evaluated at 52-weeks via telephone follow-up. The primary endpoint in the trial is 26 weeks. Participants who miss a followup will be contacted via email and telephone, and primary outcomes will be obtained over the phone if there is risk of lost followup.

The primary outcome measure is activity limitation assessed using nine items from Version 2.1 of the Oswestry Disability Index [38], with the "sex life" item substituted with a question addressing "work/household activities" [39,40]. The Oswestry is a well-established tool with strong reliability, validity, and responsiveness for evaluating activity limitation in individuals with low back pain and associated leg symptoms [41,42].

The secondary outcomes are detailed in S5 File. Additionally, a summary of all patient outcomes along with their measurement timelines is provided in Table 4.

## Semi-structured interviews

Qualitative data will be collected through semi-structured interviews (conducted in English or Kannada based on participant preference) to gather in-depth insights into the participants' experiences with the physiotherapy intervention (individualised physiotherapy or usual care). A semi-structured interview schedule (S6 File) has been developed by the researchers, guided by pertinent literature [43–46]. These sessions, conducted face-to-face in private, will ensure participant comfort and confidentiality. Interviews will be conducted by researchers not involved in the treatment of the participants, and will be audio recorded with written informed consent. One of the interviewers is a physiotherapist and PhD student (MS), and the other is a physiotherapist with 4 years of experience treating CLBP. Recruitment will continue until data saturation is achieved, defined as no new themes emerging [47].

The interviewers have received training from experienced qualitative researchers (including AJH). The interviewers will promote reflexivity by keeping a research diary to document personal biases and perspectives throughout the study

**Table 4. Patient outcome measures.**

| Outcome measure | Measurement points (weeks) |
|---|---|
| **Primary outcome measure** | |
| 1. Activity limitation: Oswestry Disability Index V2.1 with "sex life" question replaced by a "work/housework" question | 0, 5, 10, 26 and 52 |
| **Secondary outcome measures** | |
| 1. Back pain intensity over the last week (0–10 numerical pain rating scale) | 0, 5, 10, 26 and 52 |
| 2. Leg pain intensity over the last week (0–10 numerical pain rating scale) | 0, 5, 10, 26 and 52 |
| 3. Brief Pain Inventory: pain severity and pain interference | 0, 5, 10, 26 |
| 4. Global rating of change scale (7-point Likert scale) | 5, 10, 26 |
| 5. Satisfaction with treatment (5-point Likert scale) | 5, 10, 26 |
| 6. Interference with work productivity in the past week (0–10 scale), and number of work hours missed in the past week | 0, 5, 10, 26 |
| 7. Örebro Musculoskeletal Pain Screening Questionnaire Short Form (ÖMPSQ-SF) | 0, 5, 10, 26 |
| 8. Quality of life (EuroQol-5D-5L) | 0, 5, 10, 26 |
| 9. Depression Anxiety and Stress Scale (DASS-21) | 0, 5, 10, 26 |
| 10. Pain Self-Efficacy questionnaire (PSEQ) | 0, 5, 10, 26 |
| 11. Pain Catastrophizing Scale (PCS) | 0, 5, 10, 26 |
| 12. Insomnia Severity Index (ISI) | 0, 5, 10, 26 |
| 13. Central Sensitisation Inventory (CSI) | 0, 5, 10, 26 |
| 14. Clinical inflammation score | 0, 5, 10, 26 |
| 15. Treatment Credibility Questionnaire | 5, 10, 26 |
| 16. Healthcare utilisation (participant diary tracking health service utilisation, imaging, medication) | 5, 10, 26 |
| 17. Resting state and task-based functional MRI (n = 30 participants) | 0, 10 |
| 18. Qualitative interviews (n = 40 participants) | 10 |

and will have regular meetings with co-researchers to discuss emerging findings. Additionally, field notes will be utilized to record non-verbal cues and contextual observations, enriching the depth of the collected data. The recordings will be transcribed verbatim, and Kannada transcripts will be translated into English for analysis. Participants will be invited to review a transcript of their interview and provide feedback or corrections.

### Adverse events

Treating physiotherapists will regularly monitor and document any negative or unwanted effects of treatment and record these in the clinical notes. During each session, physiotherapists will ask participants about their treatment experience, including any discomfort or adverse responses related to the treatment. The treatment protocol can be modified based on response to treatment. There will also be a question on all outcome questionnaires asking participants to describe any adverse or unwanted effects of treatment.

### Functional MRI study

A nested functional magnetic resonance imaging (fMRI) study will involve 30 patients with LBP (15 from the individualised physiotherapy group and 15 from the usual care group) and 10 matched healthy controls. All subjects will undergo a baseline fMRI scan, plus a follow-up scan 10 weeks later (corresponding to the end of the main intervention period in the RCT). Numerous fMRI studies have found that reversal of pain-related brain changes can be detected within this timeframe [48–53]. Healthy controls will also be rescanned after 10 weeks to control for changes potentially attributable to time or scan familiarity.

All participants will undergo both resting-state (minimum 8 minutes) and task-based functional MRI scans using a 1.5T scanner. The imaging protocol includes functional and high-resolution structural (T1) sequences. The task-based scan will involve five bouts of active straight-leg raises (30 seconds elevation followed by 30 seconds rest) on the most symptomatic or dominant side, with auditory cues to standardise timing.

Healthy controls will be carefully matched to LBP patients based on age, sex, BMI, and general activity level. Eligibility criteria for healthy controls will include: no history of LBP or sciatica (lasting more than 24 hours) at any point in life, and no musculoskeletal conditions in the past year that lasted for one week or more *or* that required medical consultation. Further details regarding the imaging parameters and participant preparation are provided in S7 File.

### Treatment adherence

Physiotherapists will document the total number of treatment sessions attended by each patient. At every session, they will assess the patient's adherence to the treatment plan through direct questioning and by checking the completed exercise charts. Based on this review, the physiotherapist will rate the patient's level of compliance on a scale from 1 to 10. Participant self-reported compliance will also be assessed with a question on each follow-up questionnaire: "on a scale of 0-10, how adherent have you been with following the advice and exercises recommended by your physiotherapist on average over the last month (from "not at all" to "fully")" [54].

### Data management and trial monitoring

Study personnel will complete Good Clinical Practice training. Standardised questionnaires, assessment methods and recording methods will be employed to ensure consistency of data collection for all participants. Personal or identifiable participant information will be kept in each participant's medical record at JSS Hospital. Physical data (e.g., questionnaires) will be labelled only with trial identification codes to maintain confidentiality, and will be securely kept in locked filing cabinets separately to any identifying information. All data will be entered into

password-protected computers of the researchers, along with scanned copies of paper-based questionnaires for auditing and archiving purposes. Regular random audits will be conducted to ensure accurate data entry consistent with the scanned questionnaires. Audits will also involve cross-checking and triangulation of data from different sources, such as between participant questionnaires and treating therapist clinical notes. Collated and de-identified data spreadsheets will be accessible only by the research team through a secure organisational share-drive. Trial monitoring (e.g., recruitment and follow-up rates, review of compliance, review of any reported adverse events) will be completed at fortnightly meetings involving the research team members. As a low-risk study where only common physiotherapy treatment modalities will be delivered, a data monitoring committee will not be necessary, but one will be established if adverse events or participant complaints require a decision around trial continuance. The trial commenced recruiting participants in August 2024, with the first participant enrolled on 23rd August 2024. Recruitment is ongoing, and is anticipated to be completed by December 2025. Data collection is anticipated to then be completed in December 2026, with data analysis and dissemination of findings planned for 2027. Results will be reported on the trial registry, as well as in journal articles and at conferences.

## Sample size

This superiority RCT will test the null hypothesis that there is no difference in Oswestry scores between individualised physiotherapy and usual physiotherapy care. The total sample size required to detect a standardised mean difference of at least 0.5 between the two groups on the Oswestry, using a two-tailed test with a significance level ($\alpha$) of 0.05, statistical power of 0.80, and an equal group allocation (1:1), was calculated at 128 participants [55]. To accommodate a potential attrition rate of up to 20%, the sample size was adjusted to 154 participants (77 per group). An effect size of 0.5 (Cohen's d) aligns with the findings from the previous STOPS trial [13] and is generally recognised as a key threshold representing clinical relevance [56].

## Data analysis

Data will be analysed using intention-to-treat principles [57], using IBM SPSS Version 30.0 or later (SPSS Inc., Chicago, IL, USA) and Microsoft 365 Excel (Microsoft Corp., Redmond, WA, USA).

  **Analysis of clinical outcomes (treatment efficacy).**  A summary of patient characteristics at baseline will be presented using descriptive statistics. Analysis will determine between-group differences at each follow-up point (5, 10, 26 and 52 weeks), with associated 95% confidence intervals. Continuous outcome variables will be analysed using linear mixed-effects models, incorporating group-by-time interactions and modelling time as a repeated measure, while adjusting for baseline scores. Key assumptions for using linear mixed-effects models including linearity, normally distributed residuals, and homoscedasticity, will be evaluated via residual plots (Q-Q plots, histograms) and scatter plots. Where assumptions are violated, data transformation methods (e.g., log transformation) will be attempted, and if unsuccessful, non-parametric analysis methods will be used.

  For ordinal outcomes (e.g., Global Rating of Change, Satisfaction with treatment), the Mann-Whitney U test will be used to determine whether there are statistically significant differences in medians between groups. Participants will also be classified based on whether they achieve the minimal clinically important difference (MCID) on relevant outcome measures. MCID thresholds will be defined as follows: a 10-point change on the Oswestry Disability Index (scale of 0–100), a 2-point reduction on the 0–10 Numerical Rating Scale (NRS) pain scale, a self-rating of at least 'much improved' on the global rating of change scale, and a response of 'very satisfied' on the treatment satisfaction measure. Since these thresholds may underestimate meaningful change in some cases, additional analyses will be conducted using a higher threshold—specifically, a 50% reduction in Oswestry and Numerical Rating Scale pain scores [12,58]. Differences in the proportion of participants in each group achieving these clinical importance thresholds will be presented using risk ratios

and the number needed to treat (with associated 95% confidence intervals). Missing data will be addressed using maximum likelihood estimation within the linear mixed models [59], consistent with an intention-to-treat approach.

**Effect modifiers and mediators.** Analyses of treatment effect modifiers will be conducted to identify baseline participant characteristics associated with greater or lesser benefit from individualised physiotherapy compared to usual care. Using linear mixed effects models (IBM SPSS), effect modification will be explored using a three-way interaction term (predictor x group x time). The following potential effect modifiers have been pre-specified for evaluation:

i)  Pain type (nociceptive, neuropathic or nociplastic)

ii)  Disability score (Oswestry)

iii) Prognosis (Orebro)

iv) Duration of pain

Mediation analyses will be conducted to explore the possible mechanisms through which individualised physiotherapy exerts its effects. Linear regression models will be fitted using IBM SPSS to determine direct versus indirect (via a mediator variable) effects contributing to the total treatment effect. The following mediators have been identified a-priori for investigation:

i)  Whether self-efficacy serves as a mediating factor in the relationship between treatment (individualised physiotherapy versus usual care) and changes in disability levels as measured by the Oswestry Disability Index.

ii)  Whether clinical signs of inflammation, as measured by the clinical inflammation scale [30], mediate the relationship between treatment (individualised physiotherapy versus usual care) and outcomes related to pain severity (Numerical Rating Scale) and disability (Oswestry Disability Index).

Following testing of pre-planned effect modifier and mediation analyses, a data-driven approach will be employed to identify new hypotheses in an exploratory analysis.

**Analysis of cost-effectiveness.** Healthcare costs, including prescribed treatments and any additional interventions for each participant, will be recorded from the date of first enrolment up to a 6-month follow-up. Healthcare resource utilization will be collected using standardized follow-up questionnaires completed by participants at 5, 10, and 26 weeks post-enrolment. A within-study cost-utility analysis will be conducted from a healthcare perspective, with all costs valued based on Tier 2 city charges (specifically for Mysuru, classified as a Tier 2 city by the Government of India), with all costs reported in Rupees. Costs associated with time off work will be analysed separately to healthcare costs.

All analyses will be conducted using the intention-to-treat approach, with imputation methods applied to address missing data. For partially missing information related to healthcare resource utilization, relevant averages will be used for replacement. For example, if a patient reports using ibuprofen but does not specify the dosage, the average ibuprofen dosage from other patients in the same treatment group at that time point will be imputed. For fully missing data, such as instances where patients did not return a questionnaire, missed an item on the EuroQol, or failed to answer a complete question, multiple imputation will be employed, using five imputed datasets [60,61]. Between-group differences in costs will be calculated via linear mixed models.

For health outcomes, each patient's time-weighted average EuroQol utility score across the 6-month study period will be calculated via the area under the curve method [62]. This will yield health outcomes expressed in quality-adjusted life years (QALYs), which indicates the cost to gain one additional year of life spent in perfect health [63]. The mean between-group difference in QALYs will then be derived from a linear mixed model adjusting for baseline scores, using data sets obtained following multiple imputation of missing values. These analyses will provide incremental costs and incremental

 

health benefits. The incremental cost-effectiveness ratio (ICER) will then be derived by dividing the incremental cost by the incremental QALYs, which indicates the cost to gain one additional QALY [60,61].

To assess uncertainty around the ICER, nonparametric bootstrapping will be applied using a customized Microsoft Excel spreadsheet [64]. This will involve generating 5000 randomly resampled data sets (1000 samples for each of the five imputed data sets). Each of the 5000 bootstrapped cost-utility pairs will then be graphed on the cost-effectiveness plane [63,65,66]. A cost-effectiveness acceptability curve will then be generated to determine the probability that individualised physiotherapy is cost-effective compared with usual care, at different willingness to pay thresholds.

**Analysis of qualitative data.** Data analysis will follow the principles of Interpretative Phenomenological Analysis (IPA) [67], involving iterative and detailed examination of the transcripts. Two researchers (MS and AJH) will independently perform line-by-line coding. Recurrent themes across participants will be identified and categorised. The researchers will then meet to discuss their analyses along with field notes and contextual observations to reach a consensus on major themes and sub-themes. NVivo 14.0 software will be used to assist in organizing the data.

The interviews will be coded by researchers MS and AJH. MS is an Indian musculoskeletal physiotherapist with 10 years of experience in treating patients with chronic low back pain (CLBP). The other coder (AJH), is a physiotherapists with a PhD, bringing more than 20 years of clinical and academic experience in the LBP field.

**Analysis of fMRI data.** Data will be pre-processed and analyzed using SPM12 (the welcome department of cognitive neurology, London, UK, http://www.fil.ion.ucl.ac.uk/spm/software/spm12/). The standard conventional preprocessing pipeline will be adopted as outlined in the CONN-fMRI functional connectivity toolbox (http://www.nitrc.org/projects/conn).

All resting state functional images will be slice-time corrected and realigned to the first volume using a six-parameter rigid body transformation. The anatomical image and functional images will be co-registered for the corresponding time-point. Segmented grey matter and white matter images of all participants will be used to construct tissue probability maps. The template will be normalized to Montreal Neurological Institute (MNI). The smoothing kernel for the functional images will be kept as 6 mm and 2 mm for the anatomical image.

Functional connectivity analyses will be carried out using the CONN-fMRI functional connectivity toolbox v14. Seed-to-voxel and/or ROI-to-ROI functional connectivity maps will be created for each participant. The whole brain voxel to voxel/ ROI-to-ROI analysis will be used to identify possible differences between before and after treatment. For this analysis we will use all the provided areas. The mean BOLD time series will be computed across all voxels within each ROI.

Individual seed-to-voxel and ROI-to-ROI maps will be entered into a second-level analysis. A within group ROI-to-ROI analysis will be performed. Seed-to-voxel analyses will be performed if necessary.

Statistical analyses will employ linear mixed models to detect differences between healthy controls and CLBP participants at baseline and follow-up, and between treatment groups (individualised physiotherapy versus usual care) at follow-up.

## Discussion

This randomised controlled trial will determine the effectiveness of the STOPS approach to individualised physiotherapy versus usual physiotherapy care for people with CLBP in India. The trial will also explore cost-effectiveness, patient perspectives, effect modifiers, mediators and brain-related changes in pain processing. This comprehensive evaluation will determine whether the STOPS approach to individualised physiotherapy has potential to improve outcomes for people with CLBP in India.

A limitation of the trial is the implementation of the STOPS approach within one centre in India. This restricts the generalisability of findings to a limited participant population, therapist pool, and treatment setting. However, the trial provides an opportunity to evaluate the outcomes and feasibility of implementation before dedicating additional resources to wider implementation across multiple centres in India.

## Supporting information

**S1 File. Protocol submitted for ethical approval.**
(DOCX)

**S2 File. SPIRIT 2025 checklist.**
(DOCX)

**S3 File. Description of the STOPS subgroups.**
(DOCX)

**S4 File. Detailed description of STOPS treatment strategies.**
(DOCX)

**S5 File. Description of secondary outcome measures.**
(DOCX)

**S6 File. Interview guide for patients.**
(DOCX)

**S7 File. Procedures for fMRI.**
(DOCX)

## Author contributions

**Conceptualization:** Andrew J Hahne, Mohini Shah, Muhammed Rashid, Kavitha Raja, Musa Sani Danazumi, Jon Ford.

**Data curation:** Kavitha Raja.

**Funding acquisition:** Kavitha Raja.

**Investigation:** Kavitha Raja.

**Methodology:** Andrew J Hahne, Mohini Shah, Muhammed Rashid, Kavitha Raja, Musa Sani Danazumi, Jon Ford.

**Project administration:** Mohini Shah, Muhammed Rashid, Kavitha Raja.

**Resources:** Kavitha Raja.

**Supervision:** Andrew J Hahne, Mohini Shah, Muhammed Rashid, Kavitha Raja, Jon Ford.

**Writing – original draft:** Andrew J Hahne, Mohini Shah, Muhammed Rashid, Musa Sani Danazumi.

**Writing – review & editing:** Andrew J Hahne, Mohini Shah, Muhammed Rashid, Kavitha Raja, Musa Sani Danazumi, Jon Ford.

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
