## [Decision Letter · Decision Letter 0]

23 Oct 2025

Dear Dr. Hahne 

Thank you for submitting your manuscript to PLOS ONE. After careful consideration, we feel that it has merit but does not fully meet PLOS ONE’s publication criteria as it currently stands. Therefore, we invite you to submit a revised version of the manuscript that addresses the points raised during the review process.

Please submit your revised manuscript by Dec 07 2025 11:59PM. If you will need more time than this to complete your revisions, please reply to this message or contact the journal office at plosone@plos.org . A rebuttal letter that responds to each point raised by the academic editor and reviewer(s). You should upload this letter as a separate file labeled 'Response to Reviewers'.A marked-up copy of your manuscript that highlights changes made to the original version. You should upload this as a separate file labeled 'Revised Manuscript with Track Changes'.An unmarked version of your revised paper without tracked changes. You should upload this as a separate file labeled 'Manuscript'.

We look forward to receiving your revised manuscript.

Kind regards,

Mansour Abdullah Alshehri

Academic Editor

PLOS ONE

Journal Requirements:

https://www.sciencedirect.com/science/article/abs/pii/S1551714425001545?via%3Dihub

In your revision ensure you cite all your sources (including your own works), and quote or rephrase any duplicated text outside the methods section. Further consideration is dependent on these concerns being addressed.

“JSS College of Physiotherapy funded the fMRI scans and physiotherapy treatment.”

4. Please note that funding information should not appear in the Acknowledgments section or other areas of your manuscript. We will only publish funding information present in the Funding Statement section of the online submission form. Please remove any funding-related text from the manuscript. 

Reviewers' comments:

Reviewer's Responses to Questions

**Comments to the Author**

1. Does the manuscript provide a valid rationale for the proposed study, with clearly identified and justified research questions?

Reviewer #1: Yes

Reviewer #2: Yes

2. Is the protocol technically sound and planned in a manner that will lead to a meaningful outcome and allow testing the stated hypotheses?

Reviewer #1: Yes

Reviewer #2: Yes

3. Is the methodology feasible and described in sufficient detail to allow the work to be replicable?

Reviewer #1: Yes

Reviewer #2: Yes

4. Have the authors described where all data underlying the findings will be made available when the study is complete?

Reviewer #1: Yes

Reviewer #2: Yes

5. Is the manuscript presented in an intelligible fashion and written in standard English?

Reviewer #1: Yes

Reviewer #2: Yes

You may also provide optional suggestions and comments to authors that they might find helpful in planning their study.

Reviewer #1: Comments

General comments: This is a well-written protocol that largely follows rigorous statistical analytic techniques for assessing the effectiveness of STOPS. The cost-effectiveness analysis and qualitative aspect are quite detailed and well written. Here are a few comments

1. State the type of trial being conducted (superiority, non-inferiority, equivalence)

2. Clearly state the hypothesis the hypothesis to be tested based on the type of trial being conducted

3. The authors indicated that all participants would receive 10 x 45-minute physiotherapy sessions throughout 10 weeks, but did not provide any justification for the intervention period and its implications. Why 10 weeks? For instance, does extending the period beyond 10 weeks change the effect of the intervention (positive or negative)? What would have been the implication (positive or negative) if a shorter duration had been chosen? Details must be provided

4. Separate the power analysis (sample size calculations) from the data analysis section. There should be separate subheadings for sample size (power analysis) and statistical analysis to aid the flow of the protocol. The sample size estimation must be based on a clearly defined hypothesis and the type of RCT, as stated previously. There should be a section on data quality assurance for the trial.

5. Kindly include the key assumptions under linear mixed-effects models and how these assumptions will be tested. Include regression modelling strategies that will be adopted if these assumptions are violated

6. The authors stated that “For ordinal outcomes, the Mann-Whitney U test will be used,” but did not indicate what the test would be used for. The statement is not complete

7. The statement “Dichotomous outcomes will be presented using risk ratios and the number needed to treat (with associated 95% confidence intervals), with significance determined using chi-square (χ²) analyses.” is unclear and somewhat confusing. Please revise and specify when you will use the Chi-square test and break sentences to make it a bit clearer.

8. The authors did not mention the statistical tool for mediation analysis. This must be included in the manuscript.

9. The authors did not include the statistical tool for assessing effect modification. Thy only mentioned the effect modifiers. This must be included in the manuscript.

Reviewer #2: 1.In exclusion criteria they mentioned minimal physical activity limitation will be excluded. From my point there is no need to mention it rather than mention the minimal physical activity limitation criteria like can sit more than 1 hour without pain etc.

2. Usual Care: Description of usual care should be included

3. Who will be the treatment provider: You mentioned only 2 PT trained about STOP approach. Is it possible to serve only by them.

**Do you want your identity to be public for this peer review?** For information about this choice, including consent withdrawal, please see our Privacy Policy

Reviewer #1: No

Reviewer #2: **Yes: ** Md. Zahid Hossain

---

## [Author Response · Author response to Decision Letter 1]

5 Nov 2025

Thank-you for the opportunity to resubmit a review of this paper. The helpful comments of the editor and reviewers have been addressed in the revised manuscript, and a point-by-point response to reviewers has been uploaded in the files.

---

## [Editor Report · Decision Letter 1]

4 Dec 2025

STOPS approach to individualised physiotherapy versus usual physiotherapy care for chronic low back pain in India: A randomised controlled trial protocol

PONE-D-25-44126R1

Dear Dr. Hahne,

We’re pleased to inform you that your manuscript has been judged scientifically suitable for publication and will be formally accepted for publication once it meets all outstanding technical requirements.

Kind regards,

Mansour Abdullah Alshehri

Academic Editor

PLOS ONE
---

## [Editor Report · Acceptance letter]

PONE-D-25-44126R1

PLOS One

Dear Dr. Hahne,

I'm pleased to inform you that your manuscript has been deemed suitable for publication in PLOS One. Congratulations! Your manuscript is now being handed over to our production team.

Kind regards,

on behalf of

Dr. Mansour Abdullah Alshehri

Academic Editor

PLOS One